# Projection of snowfall extremes in the French Alps as a function of elevation and global warming level

Erwan Le Roux[1], Guillaume Evin[1], Raphaëlle Samacoïts[2], Nicolas Eckert[1], Juliette Blanchet[1], and Samuel Morin[2]

[1]Univ. Grenoble Alpes, INRAE, CNRS, IRD, Grenoble INP, IGE, 38000 Grenoble, France
[2]Univ. Grenoble Alpes, Univ. Toulouse, Météo France, CNRS, CNRM, CEN, Grenoble, France

**Correspondence:** Erwan Le Roux (erwan.le-roux@univ-grenoble-alpes.fr)

**Abstract.** Following the projected increase in extreme precipitation, an increase in extreme snowfall may be expected in cold regions, e.g. for high latitudes or at high elevations. By contrast, in low/medium elevation areas, the probability of experiencing rainfall instead of snowfall is generally projected to increase due to warming conditions. Yet, in mountainous areas, despite the likely existence of these contrasted trends according to elevation, changes in extreme snowfall with warming remain poorly

quantified. This paper assesses projected changes in heavy and extreme snowfall, i.e. in mean annual maxima and 100-year return levels, in the French Alps as a function of elevation and global warming level. We apply a recent methodology, based on the analysis of annual maxima with non-stationary extreme value models, to an ensemble of 20 adjusted GCM-RCM pairs from the EURO-CORDEX experiment under the RCP8.5 scenario. For each of the 23 massifs of the French Alps, maxima in the hydrological sense (August 1st to July 31st) are provided from 1951 to 2100, and every 300 m of elevations between 900 m

and 3600 m. Results rely on relative or absolute changes computed with respect to current climate conditions (corresponding here to +1°C global warming level), at the massif scale and averaged over all massifs. Overall, daily mean annual maxima of snowfall are projected to decrease below 3000 m and increase above 3600 m, while 100-year return levels are projected to decrease below 2400 m and increase above 3300 m. At elevations in between, values are on average projected to increase until +3°C of global warming, and then decrease. At +4°C, average relative changes in mean annual maxima and 100-year return

levels respectively vary from −26% and −15% at 900 m, to +3% and +8% at 3600 m. Finally, for each global warming level between +1.5°C and +4°C, we compute the elevation threshold that separates contrasted trends, i.e. where the average relative change equals zero. This elevation threshold is shown to be lower for higher return periods, and is projected to rise from 3000 m at +1.5°C to 3350 m at +4°C for mean annual maxima, and from 2600 m to 3000 m for 100-year return levels. These results have implications for the management of risks related to extreme snowfall.

## 1 Introduction

Extreme snowfall can cause major natural hazards (avalanche, winter storms, snow loads) which may generate casualties and economic damage (Changnon, 2007; Blanchet et al., 2009; Le Roux et al., 2020). Despite these strong implications for societies, to which extent ongoing climate change affects extreme snowfall remains poorly quantified (IPCC, 2019, 2021).

The two main physical drivers of extreme snowfall, temperature and extreme precipitation, are both expected to increase with anthropogenic climate change (IPCC, 2021). Following the increase of global mean temperatures, temperatures are expected to increase more over lands than over oceans (Byrne and O'Gorman, 2018). The warming rate at higher elevations can either be amplified or show no significant difference when compared with the warming in lowland regions (Pepin et al., 2015, 2022). At the global scale, extreme precipitation is projected to increase of 7% per °C of global mean warming, due to an increase in mean atmospheric water vapor content according to the Clausius–Clapeyron relationship (Ingram, 2016; Allan et al., 2020). At the regional scale, we note that changes in atmospheric circulation patterns might modulate these warming-induced trends (Frei et al., 2018; Blanchet et al., 2020, 2021). In particular, relative changes in extreme precipitation per unit of local warming are not evenly distributed over the globe and reach lower rates over land (see, e.g., Fig. 6 in Kharin et al. 2013).

Contrasted trends in extreme snowfall are expected (O'Gorman, 2014). A decrease is projected in low/medium elevation areas due to warming conditions. In cold regions, e.g. for high latitudes or at high elevations, one can expect an increase in extreme snowfall. Many recent studies based on climate projections illustrate this phenomenon with maps of extreme snowfall trends (Lader et al., 2020; Chen et al., 2020; Kawase et al., 2021; Quante et al., 2021). For instance, Lader et al. (2020) found that maximum 2-day snowfall projections in Alaska either show no trends or decreasing trends with the RCP8.5 scenario. On the other hand, increasing trends in annual maximum and monthly maximum daily snowfall have been found over several regions in Japan with the RCP8.5 scenario (see Fig. S3 in Kawase et al. 2021). In the Northern Hemisphere, using CMIP6 simulations under the SSP5-8.5 scenario, Chen et al. (2020) find significant increasing trends in high-latitude regions, and Quante et al. (2021) show an intensification of extreme snowfall across large areas. Recently, using CMIP6 simulations and reanalysis, Ombadi et al. (2023) show that rainfall extremes are amplified in high-elevation regions of the Northern Hemisphere due to a warming-induced shift from snow to rain.

Regionally, it is sometimes possible to identify a threshold that separates contrasted trends. For instance, in the French Alps between 1959 and 2019, Le Roux et al. (2021) find an elevation threshold that is roughly 2000 m, i.e. that extreme snowfall has decreased below 2000 m and increased above 2000 m over the study period. Table 1 reports studies based on climate projections that identify a threshold below or above which extreme snowfall is projected to decrease or increase, respectively. This threshold can be specified either in terms of elevations (López-Moreno et al., 2011; Frei et al., 2018) or with climatological temperatures (de Vries et al., 2014; Lute et al., 2015; Kawase et al., 2016). Studies that directly estimate a threshold usually compute it for fixed time periods, e.g. 2070-2100, and for low return periods (mean annual maxima or 99th percentile of daily values), even though we note the exception of Kawase et al. (2016) and López-Moreno et al. (2011) that study larger 10- and 25-year return levels.

Our study assesses projected changes in heavy (mean annual maxima) and extreme (100-year return level) snowfall in the French Alps under a high emission scenario (RCP8.5), primarily on daily values. We focus on the 100-year return level because the corresponding return period is often considered to mitigate snow-related hazard study (Gaume et al., 2013). Changes are estimated as a function of global warming level using a recent methodology (Le Roux et al., 2022), based on non-stationary extreme value analysis and designed for climate projection ensembles. Hence, we provide new projected trends in extreme snowfall as a function of global warming levels in the French Alp. We also provide the evolution of the elevation threshold

| Reference | Location | Indicator | Projected changes | Periods | Dataset | Scenario |
|---|---|---|---|---|---|---|
| López-Moreno et al. (2011) | Pyrenees | 25-year return level | Decrease below 1500 m<br>Increase above 2500 m | 1960-1990 vs 2070-2100 | 1 RCM,<br>55 km resolution | SRES A2 |
| de Vries et al. (2014) | Western Europe | Dec-Feb mean annual maximum | Decrease for $T^{\text{histo}}_{\text{Dec-Feb}} > -5^oC$<br>Increase for $T^{\text{histo}}_{\text{Dec-Feb}} < -10^oC$ | 1981-2010 vs 2071-2100 | 1 RCM,<br>12 km resolution | RCP8.5 |
| Lute et al. (2015) | Western USA | Average above 99th percentile | Decrease for $T^{\text{histo}}_{\text{Nov-Mar}} > -3^oC$<br>Increase for $T^{\text{histo}}_{\text{Nov-Mar}} < -7^oC$ | 1950-2005 vs 2040-2069 | 20 GCMs downscaled at stations | RCP8.5 |
| Kawase et al. (2016) | Japan and North Asia | 10-year return level Nov-Apr | Decrease for $T^{\text{histo}}_{\text{Nov-Mar}} > -5^oC$<br>Increase for $T^{\text{histo}}_{\text{Nov-Mar}} < -5^oC$ | 1950-2011 vs 2080-2099 | 1 GCM 48 runs,<br>20 km resolution | RCP8.5 |
| Frei et al. (2018) | European Alps | Sep-May mean annual maximum | Decrease up to 3000 m a.s.l<br>Increase above 3000 m a.s.l | 1981-2010 vs 2070–2099 | 14 GCM-RCMs,<br>12 km resolution | RCP8.5 |

**Table 1.** Projected changes in extreme snowfall under a high emission scenario. Based on climate projection datasets, changes are assessed between an historical and a future period. $T^{\text{histo}}_{\text{m1-m2}}$ denotes the mean temperature for the historical period between the months m1 and m2, which can be used as a threshold to define regions where extreme snowfall are expected to increase or to decrease.

below or above which extreme snowfall is projected to decrease or increase, respectively, on average in the French Alps. To the best of our knowledge (Tab. 1), this study is the first to report on the computation of such a threshold for high return periods and as a function of global warming level.

## 2 Data

Annual maxima of daily snowfall are provided for the 23 massifs of the French Alps (Fig. 1a) by the S2M reanalysis (Durand et al., 2009a; Vernay et al., 2019, 2022) which combines large-scale reanalyses, meteorological forecasts and ground measurements. These reanalyses of daily snowfall data, expressed in kg m$^{-2}$, span the time period from August 1958 to July 2019. Here, for any year $t$ between 1959 and 2019, we consider the annual maxima for the period between the 1st of August for the year $t-1$ and the 31st of July for the year $t$. By construction, the S2M reanalysis introduces a relationship between the elevation and the meteorological conditions and directly provides meteorological variables every 300 m of elevation between 900 and 3600 m, for each massif (for some massifs, the elevation range is restrained, according to their topography). In this article, we focus on maxima of daily snowfall values, but we also perform an analysis based on maxima of 3- and 5-days snowfall values (Supplement, Part E).

Climate simulations of annual maxima of daily snowfall are obtained for an ensemble of 20 regional simulations obtained with 6 CMIP5 GCMs and 11 RCMs (see Supplement, Part A). Annual maxima are available both for the historical emission scenario accounting for anthropogenic and natural radiative forcing (1951–2005) and for the high emission scenario RCP8.5. (2006–2100). These annual maxima of daily snowfall are computed in three steps. First, simulations of daily precipitation from the EURO-CORDEX project (Jacob et al., 2014) are downscaled and corrected at the massif-level and every 300 m of

elevation using an advanced quantile-mapping method (so-called ADAMONT, see Verfaillie et al., 2017). The ADAMONT method relies on the S2M reanalysis as a reference and is applied separately for the four seasons, four weather regimes, and several meteorological variables including temperature and precipitation. Then, daily precipitation is disaggregated at the hourly time step using analogues and partitioned between rain and snow with the threshold $1^oC$ and an additional quantile-mapping correction is applied (see Verfaillie et al., 2017, for details). Finally, daily snowfall data are computed by aggregating hourly snowfall and the annual maxima of the daily snowfall values are computed (annual values taken from August 1st to July 31st, so as to cover the hydrological year). For 9 massifs at 900 m and 3 massifs at 1200 m, some projected annual maxima are equal to zero. In order to avoid the estimation and statistical treatment of this zero-snowfall probability, these simulations are discarded from the analysis (see section 5.1 in Le Roux et al., 2021, for further discussion). Figure 1b provides an illustration of past and projected time series of annual maxima for the Vanoise massif at 1500 m elevation.

Following the methodology proposed by Le Roux et al. (2022), the temporal covariate chosen in this study is the smoothed anomaly of global mean surface temperature (GMST) with respect to the pre-industrial period (1850–1900). This smoothed anomaly is obtained with cubic splines. Thus, even if our analysis is conditional on the use of the RCP8.5 emission scenario, using global warming level as covariate makes our approach more universal than in a time-dependent approach using this sole scenario. For the reference data, i.e. the S2M reanalysis, we exploit the GMST obtained from the HadCRUT5 reanalysis (Morice et al., 2021) while for each GCM–RCM pair, the chosen GMST is obtained from the corresponding large-scale GCM simulation (see Fig. 1c). Hereafter, the smoothed anomaly of GMST of $+1°C$ is referred to as the "current climate" since this level of warming roughly corresponds to the current climate conditions (IPCC, 2021).

## 3  Methodology

Let $Y_t^{\text{obs}}$ denote an annual maximum from the S2M reanalysis (Sect. 2) for the year $t$ between 1959 and 2019, and $T_t^{\text{obs}}$ represent the smoothed anomaly of global mean surface temperature (GMST) from HadCRUT5 for the same year $t$. For a GCM-RCM pair $k$, let $Y_t^k$ represent an annual maximum for the year $t$ between 1951 and 2100, and $T_t^k$ represent the smoothed anomaly of GMST for the corresponding GCM. We apply a recent statistical methodology (Le Roux et al., 2022) that relies on a non-stationary Generalized Extreme Value (GEV) model (Coles, 2001, and references therein) that combines past reference and a climate projection ensemble. In summary, the GEV model of Le Roux et al. (2022) includes a possible evolution of extreme value distribution by considering piecewise-linear functions $\mu(.), \sigma(.), \xi(.)$ for each of the three GEV parameter:

$$Y_t^{\text{obs}}|\boldsymbol{\theta} \sim \text{GEV}(\mu(T_t^{\text{obs}}), \sigma(T_t^{\text{obs}}), \xi(T_t^{\text{obs}})) \quad \text{with} \quad \begin{aligned} \mu(T) &= \mu_0 + \sum_{i=1}^{L} \mu_i \times (T - \kappa_i)_+, \\ \log \sigma(T) &= \sigma_0 + \sum_{i=1}^{L} \sigma_i \times (T - \kappa_i)_+, \\ \xi(T) &= \xi_0 + \sum_{i=1}^{L} \xi_i \times (T - \kappa_i)_+, \end{aligned} \quad (1)$$

where $\boldsymbol{\theta}$ is the vector of parameters $\{\mu_i, \sigma_i, \xi_i, i = 0, \ldots, L\}$ for the piecewise-linear functions $\mu(.), \sigma(.), \xi(.), 1 \leq L \leq 4$ corresponds to the number of linear pieces, $\kappa_i = T_{\min} + \frac{(i-1) \times (T_{\max} - T_{\min})}{L}$ for $i \in \{1, \ldots, L\}$, and $T_{\min}$ and $T_{\max}$ are the minimum and maximum smoothed anomaly of GMST for the period 1951-2100. Similarly, for the projected annual maxima, $Y_t^k$ is modelled

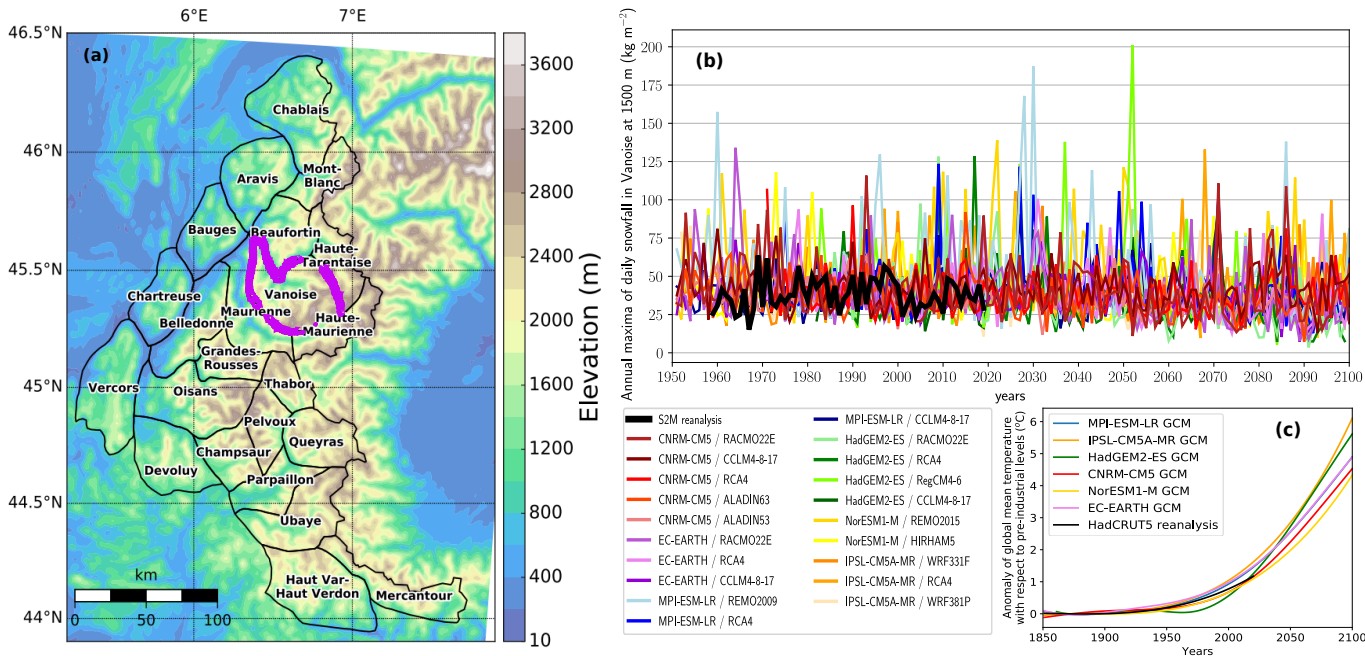

**Figure 1. (a)** Topography of the 23 massifs of the French Alps, e.g., the purple region is named the Vanoise massif (Durand et al., 2009a). **(b)** Time series of annual maxima of daily snowfall from 1951 to 2100 for the Vanoise massif at 1500 m elevation. Annual maxima from the 20 adjusted GCM–RCM pairs (1951–2100) under a historical and a high-emission scenario (RCP8.5) are displayed with bright colors, while annual maxima from the S2M reanalysis (1959–2019) are displayed in black. **(c)** Smoothed anomaly of GMST with respect to pre-industrial levels (1850–1900). For the six GCMs, we rely on historical emissions until 2005 and then on projected emissions for the RCP8.5 scenario. Years correspond to periods centered on each winter (August–July). This caption is adapted from Le Roux et al. (2022).

as follows:

$$Y_t^k|\boldsymbol{\Theta} \sim \mathrm{GEV}\big(\mu(T_t^k) + \tilde{\mu}_k, \sigma(T_t^k) + \tilde{\sigma}_k, \xi(T_t^k)\big), \tag{2}$$

where $\boldsymbol{\Theta}$ denotes the set of parameters $\boldsymbol{\theta}$ and additional parameters $\tilde{\mu}_k$ and $\tilde{\sigma}_k$ which correspond to different adjustment
coefficients. These adjustment coefficients can account for systematic differences between the different climate trajectories. The number of adjustment coefficients can vary according to the selected parameterization: no adjustment coefficient, one adjustment coefficient for all GCM-RCM pairs, one for each GCM, one for each RCM, or one for each GCM-RCM pair. A two-step selection approach is applied in order to automatically choose the optimal configuration for the number of linear pieces $L$ and for the parameterization of the adjustment coefficients (Le Roux et al., 2022).

Next, we detail the three steps of our analysis: i) a non-stationary GEV model is estimated for each massif and each elevation using the reference data and the 20 GCM-RCM pairs ii) relative and absolute changes (with respect to +1°C) in mean annual maxima and 100-year return levels are assessed at the massif scale, and averaged over all the massifs for each elevation and

every 0.1 °C of global warming iii) the elevation threshold, i.e. the elevation where the average relative change is equal to $0\%$, is computed every 0.1°C of global warming for the mean annual maxima and 100-year return levels.

First, for each massif, the vector of parameters $\boldsymbol{\Theta}$ of the non-stationary GEV model is estimated using the reference data and all GCM-RCM pairs using the maximum likelihood method. This non-stationary GEV model provides a single GEV distribution of annual maxima of snowfall for each level $T$ (in °C) of global warming. Thus, for each level of global warming, mean annual maxima of snowfall can be obtained as the expectation of the GEV distribution, while the 100-year return level of snowfall corresponds to the 99th percentile of this distribution. Absolute and relative changes can be obtained for each level of global warming with reference to the GEV distribution at +1°C (which corresponds roughly to the current climate).

Then, the average relative change is defined as the relative change averaged over all available massifs of the French Alps. Massifs are considered as not available when the considered elevation is above the top elevation of the massif, or when the frequency of years without snowfall is too large (we choose to exclude massifs with a frequency above 5 %) because it might break extreme value theory assumptions. These average relative changes are computed every 300 m of elevation from 900 m to 3600 m, and every 0.1°C of global warming. Similarly, the average change is defined as the absolute change averaged over all available massifs.

Finally, for each level of global warming, we compute the elevation threshold with a contour function from the Python programming language which is based on a quadtree subdividing algorithm, see e.g. Wang and Bruch (1995). This contour function provides the elevation that corresponds to specific values of average relative change (..., $-5\%$, $0\%$, $5\%$, ...) based on average relative changes computed every 300 m of elevation from 900 m to 3600 m (Fig. 6). By definition, the elevation threshold corresponds to the elevation at the specific value of $0\%$, i.e. the elevation where the average relative change compared to our current climate (at +1°C) is equal to $0\%$. In other words, at $T$°C of global warming for all elevations above or below this threshold, extreme snowfall is projected on average to be larger or smaller, respectively, than extreme snowfall of our current climate.

## 4 Results

### 4.1 Projected changes in extreme daily snowfall at the massif level

In this subsection, we focus on a subset of four elevations (900 m, 1800 m, 2700 m, and 3600 m) out of the ten elevations considered (every 300 m from 900 m to 3600 m). Figure 2 shows the relative changes in mean annual daily maxima and 100-year return levels of snowfall between +1°C and +4°C at these four elevations. Figure 3 illustrates the corresponding absolute changes. For both indicators (mean annual maxima and 100-year return levels), a majority of massifs exhibit a decreasing trend at 900 m and 1800 m, and an increasing trend at 3600 m. At 2700 m, we observe a majority of decreasing trends for the mean annual maxima, while both decreasing and increasing trends are found for the 100-year return levels. Spatially, we observe some variability between the massifs. For the mean annual maxima, relative changes are often larger (increasing trends are greater, decreasing trends are less marked) in the Northern French Alps than in the Southern French Alps. For the 100-year return levels, we do not find any striking spatial pattern, even if we notice that relative changes are slightly larger in the

Eastern French Alps. Overall, the largest decrease is projected for the mean annual maxima of the most southeastern massif (Mercantour massif) at 1800 m: $-39\%$ ($-20$ kg m$^{-2}$). The largest decrease in 100-year return levels is expected in a northern massif (Mont-Blanc massif) at 1800 m with $-14\%$ ($-18$ kg m$^{-2}$). The largest increase is projected in a vast Northeastern massif (Vanoise massif) at 3600 m both for the mean annual maxima $+12\%$ ($+9.5$ kg m$^{-2}$) and the 100-year return levels $+13\%$ ($+22.5$ kg m$^{-2}$).

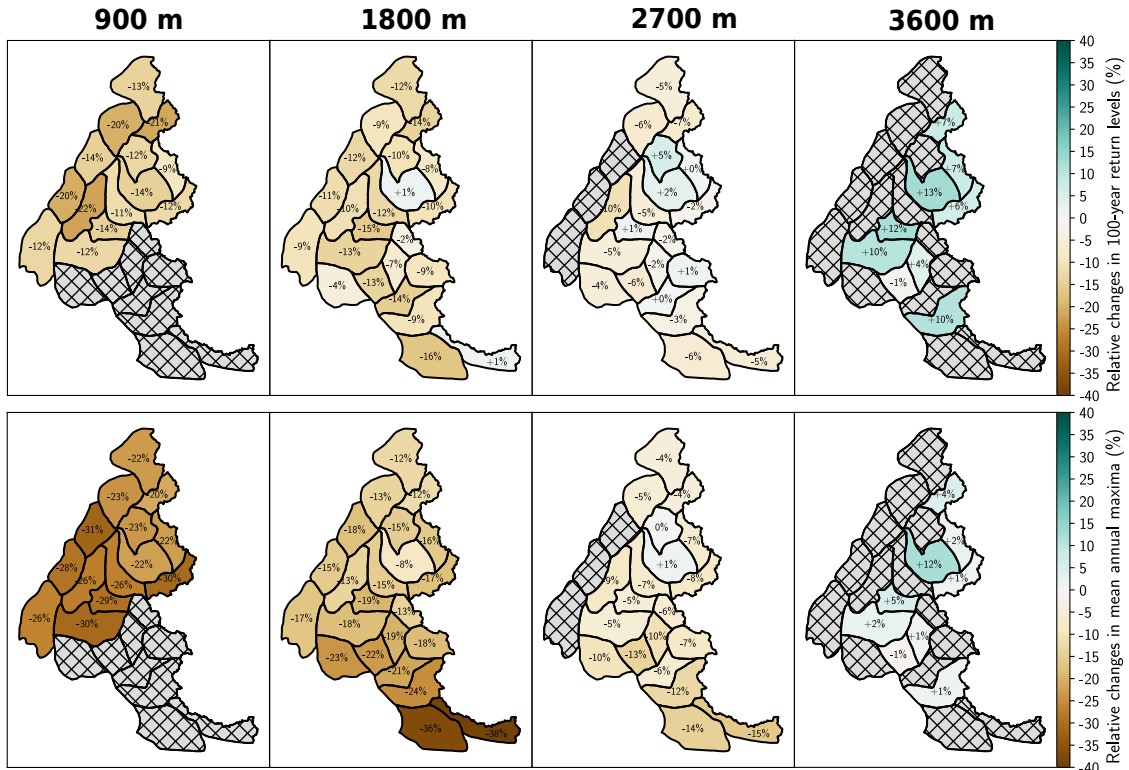

**Figure 2.** Relative changes in mean annual maxima and 100-year return levels of daily snowfall between the current climate (corresponding here to $+1°$C of global warming) and $+4°$C of global warming at elevations 900 m, 1800 m, 2700 m, and 3600 m. Hatched grey massifs denote massifs that are not available (Sect. 3, second step of our analysis).

## 4.2 Projected changes in extreme daily snowfall averaged over all available massifs of the French Alps

Figure 4 illustrates the average relative change of mean annual maxima and 100-year return levels of daily snowfall for different levels of global warming and every 300 m of elevation from 900 m to 3600 m. Figure 5 shows the corresponding average absolute changes. Mean annual maxima of daily snowfall are projected to increase at 3600 m, to slightly increase at 3300 m until $+3°$C of global warming and then to marginally decrease, and to decrease below 3000 m all over the considered warming window (Fig. 4a). Similarly, changes in 100-year return levels of snowfall are projected to increase above 3300 m, to increase until $+3°$C of global warming and then decrease at 2700 m and 3000 m, and to decrease below 2400 m (Fig. 4b). These

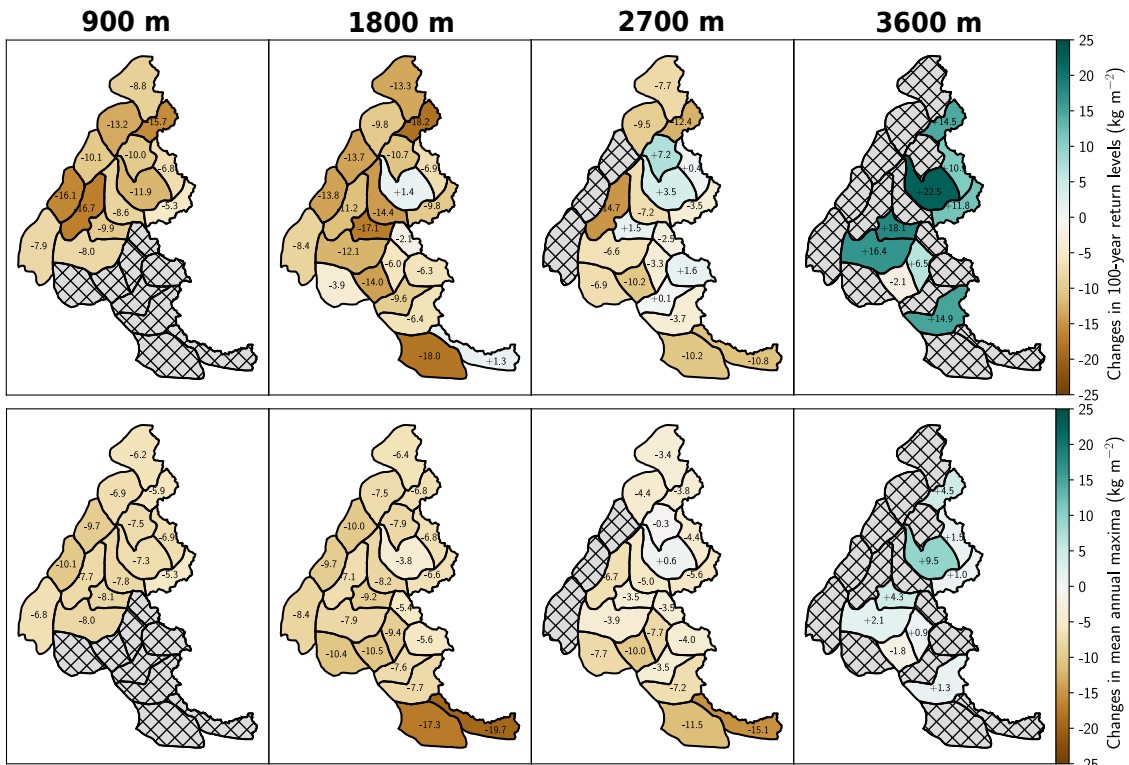

**Figure 3.** Absolute changes in mean annual maxima and 100-year return levels of daily snowfall between the current climate (corresponding here to +1°C of global warming) and +4°C of global warming at elevations 900 m, 1800 m, 2700 m, and 3600 m. Hatched grey massifs denote massifs that are not available (Sect. 3, second step of our analysis).

decreasing trends are clearly more pronounced for mean annual maxima than for 100-year return levels. Indeed, even for a global warming level of +4°C, at 900 m, 100-year return levels are projected to decrease by $-15\%$ ($-11$ kg m$^{-2}$) compared to $-26\%$ ($-8$ kg m$^{-2}$) for mean annual maxima. At +4°C of global warming, average relative changes in mean annual maxima and 100-year return levels are respectively expected to reach $+3\%$ ($+3$ kg m$^{-2}$) and $+8\%$ ($+13$ kg m$^{-2}$) at 3600 m.

In terms of goodness of fit, we find that all selected statistical models after the two-step selection approach (Sect. 3) show a relatively good fit (Supplement, Part B), and always have physically plausible shape parameters, i.e. between -0.5 and 0.5 (Martins and Stedinger, 2000, see Supplement, Part C).

We also note that similar trajectories of relative changes are observed when our statistical methodology is applied to snowfall accumulated over 3 days and snowfall accumulated over 5 days. However, overall, decreasing trends seem to be enhanced and increasing trends tempered with regards to results obtained for daily maxima (Supplement, Part E).

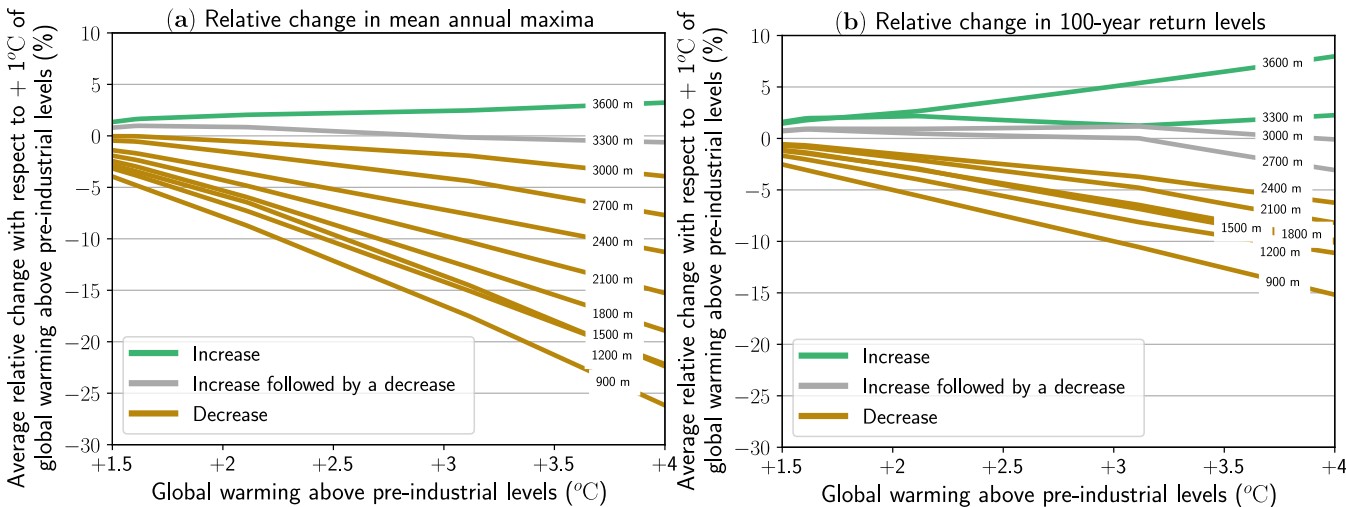

**Figure 4.** Average relative changes in **(a)** mean annual maxima **(b)** 100-year return levels of daily snowfall, every 300 m of elevation from 900 m to 3600 m, between +1.5 and +4°C global warming levels. Relative changes are computed with respect to the current climate (corresponding here to +1°C of global warming), and are averaged over all available massifs of the French Alps.

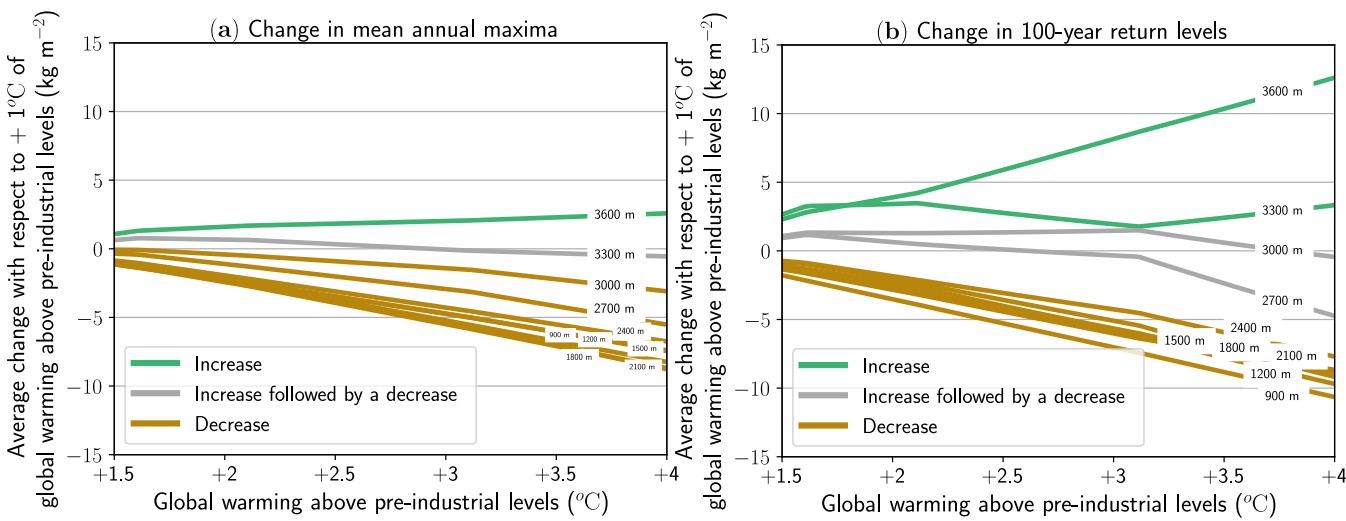

**Figure 5.** Average changes in **(a)** mean annual maxima **(b)** 100-year return levels of daily snowfall, every 300 m of elevation from 900 m to 3600 m, between +1.5 and +4°C of global warming. Changes are computed with respect to the current climate (+1°C of global warming) and are averaged over all available massifs of the French Alps.

### 4.3 Projected changes in the elevation threshold separating decreasing from increasing trends

Figure 6 illustrates the elevation that corresponds to various values (..., $-5\%$, $0\%$, $5\%$, ...) of average relative change both for the mean annual maxima and for the 100-year return levels of daily snowfall. These values are displayed between $+1.5°C$ and $+4°C$ of global warming. We observe that the elevation threshold, which corresponds to $0\%$ change, is projected to increase with global warming. For example, the elevation threshold of mean annual maxima of daily snowfall is projected to approximately increase from 3000 m at $+1.5°C$ of global warming to 3300 m at $+4°C$, whereas the elevation threshold for 100-year return levels of daily snowfall is projected to increase from 2600 m to 3000 m over the same temperature window. Figure 6 also illustrates the elevation that corresponds to other values of average relative change with respect to the current climate ($+1°C$ of global warming). For instance at 900 m, -10% is projected to be reached at roughly $+2.25°C$ of global warming for the mean annual maxima, and at $+3°C$ of global warming for the 100-year return levels.

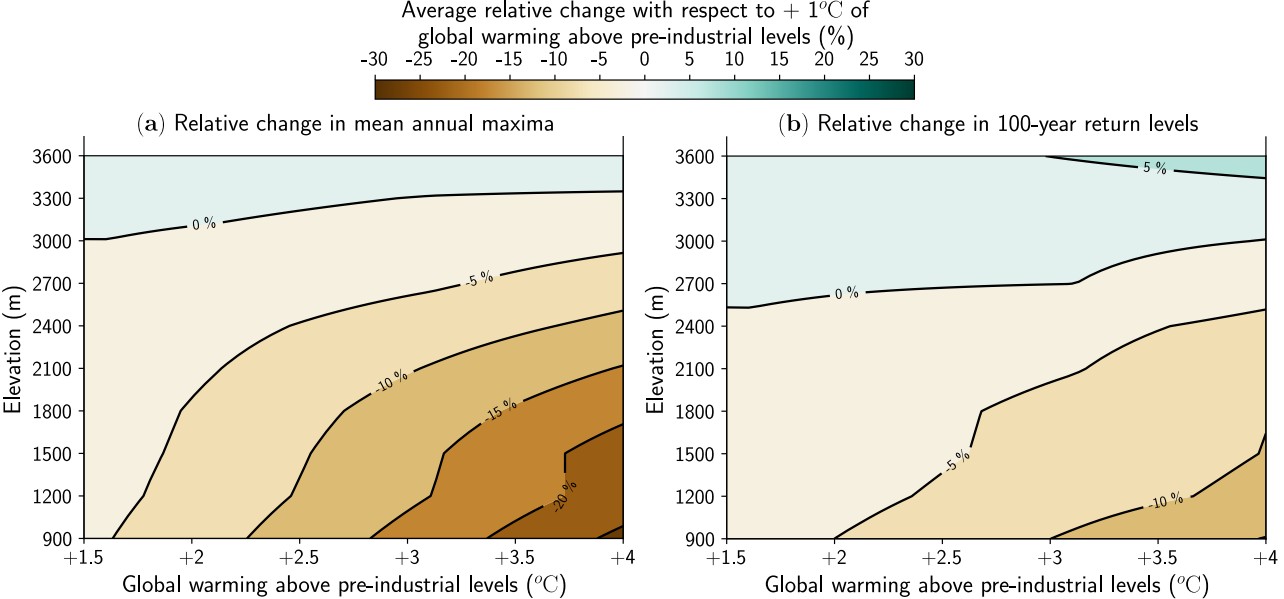

**Figure 6.** Contour plot of average relative changes in **(a)** mean annual maxima **(b)** 100-year return levels of daily snowfall. These values are shown between +1.5 and +4°C of global warming from 900 m to 3600 m. Relative changes are computed with respect to the current climate ($+1°C$ of global warming), and are averaged over all available massifs of the French Alps. The elevation threshold above or below which extreme snowfall is projected to increase or decrease, respectively, corresponds to the level $0\%$.

Figure 7 displays the elevation threshold for the mean annual maxima and for the T-year return levels with $T \in \{2, 5, 10, 20, 50, 100\}$ as a function of the global warming level. Thus, the elevation threshold of heavy snowfall, e.g. mean annual maxima and 2-year return levels, can be compared with the elevation threshold of extreme snowfall, e.g. 50-year and 100-year return levels. As shown in Figure 7, for a given warming level, the elevation threshold is always lower for higher return periods. Yet, we also find that elevation thresholds are projected to increase with warming more for extreme snowfall than for heavy snowfall. For

instance, between +1.5°C and +4°C of global warming, the elevation threshold increases at a rate of 123 m and 164 m per °C for mean annual maxima and 100-year return levels, respectively. However, the elevation threshold does not increase linearly

for the most extreme snowfall events (50-year and 100-year return levels): a steep increase is projected around +3°C of global warming which likely results, among different factors, from the statistical methodology (Sect. 5.2).

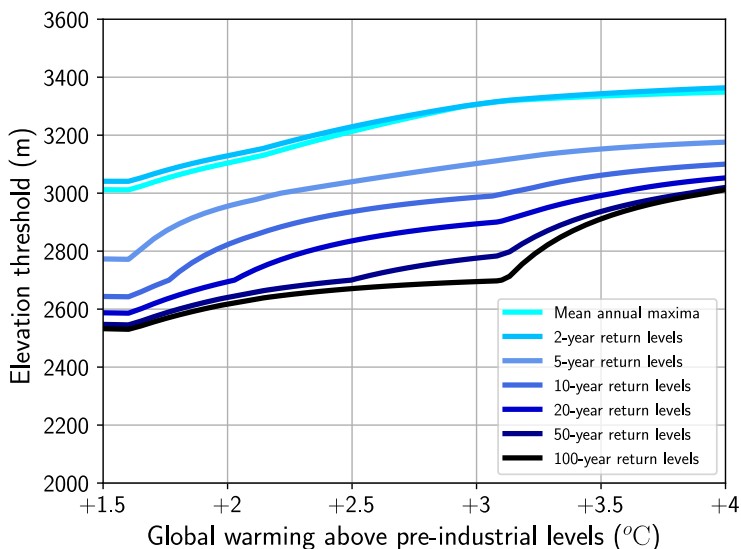

**Figure 7.** Evolution with global warming level of the elevation threshold above or below which daily extreme snowfall is projected to increase or decrease, respectively, for different return periods with respect to the current climate (+1°C of global warming).

## 5    Discussion

### 5.1    Data

In this study, the S2M reanalysis provides past snowfall reference data for the period 1959-2019 for each massif of the French
Alps and every 300 m of elevation. The S2M reanalysis assimilates many direct observations of weather and snow variables. In the Alps, more than 200 measurements stations of daily precipitation are available since the 1960s (see Vernay et al., 2022, Fig. 4b). The S2M reanalysis produces snow depth values that are consistent with snow depth observations, although strong uncertainties remain at high elevations. Above 2700 m, a few stations record surface temperatures and daily precipitation amounts. Snowfall amounts at high elevations are thus more uncertain, and an underestimation of precipitation above 3000 m
can be suspected (Vionnet et al., 2019). Annual maxima of daily snowfall are likely affected by these uncertainties and by the high spatial and temporal variability of annual maxima from daily values. Despite these known limitations, this reanalysis has

been repeatedly shown to provide valuable insights regarding past snow climate conditions and their links to snow avalanche activity in the French Alps (Durand et al., 2009b; Castebrunet et al., 2012; Schläppy et al., 2016).

The snowfall projections are obtained from 20 regional climate simulations which have been adjusted against the S2M reanalysis with the quantile mapping method ADAMONT (Verfaillie et al., 2017). Quantile mapping has known limitations: it can sometimes lead to implausible climate and it cannot overcome climate model errors (Maraun et al., 2017). However, in ADAMONT quantile mapping is applied for quantiles below 0.995: otherwise "a constant adjustment based on the value of this last quantile is applied in order to allow for new extremes". The results shown in this study are conditional to this ensemble of climate simulations obtained with different climate models (6 CMIP5 GCMs and 11 RCMs). It is also conditional on the use of the RCP8.5 greenhouse gas concentration pathway. However, the analysis in terms of global warming level makes it more universal than in a time-dependent approach using this sole scenario. One specific limitation of this work pertains to the implementation of the statistical adjustment method applied to the EURO-CORDEX datasets. Indeed, for each EURO-CORDEX model pair, a single RCM grid point is identified for each massif and the S2M data at each elevation are used to adjust the EURO-CORDEX data for this massif. Therefore, the elevation-dependence of the total precipitation trend within a given massif (Supplement, Part D provides the average behavior of intense winter precipitation for the sake of comparison with Figure 4) is mainly related to the statistical relationship between S2M and the EURO-CORDEX RCM during the adjustment period. The diverging trend on extreme solid precipitation therefore mainly stems from the combination of the trend on extreme precipitation and the effect of temperature values on the phase of the precipitation. Another limitation of this work is related to the characteristics of the GCM/RCM model pairs used in this study. The capacity of regional climate models such as those used in the EURO-CORDEX ensemble, to represent processes conducive to extreme precipitation events has been questioned in past studies, which deserves even more caution at high elevation (Rajczak and Schär, 2017). The limited ability of the RCMs from the EURO-CORDEX ensemble to simulate convection based precipitation might particularly have an impact on precipitation in the Southernmost massifs of the French Alps where convective processes play a major role. However, convective phenomena occur mostly in summertime, and most extreme precipitation events occur in wintertime throughout the mountain regions. Further studies using convection-permitting, higher resolution regional climate models, may advance this field further in the coming years, see Kotlarski et al. (2022) and references therein (Lucas-Picher et al., 2021; Ban et al., 2021; Monteiro et al., 2022). Yet, it should be noted here as well that the ADAMONT approach and its results for the considered GCM-RCM sample, while processed with suitable statistical techniques (Verfaillie et al., 2018; Evin et al., 2019), can be used to provide useful results regarding the future snow and climate conditions in the French Alps.

## 5.2 Methodology

Our study takes advantage of a recent statistical methodology that can project the evolution of any extreme variable from a climate projection ensemble (Le Roux et al., 2022). This methodology relies on flexible non-stationary generalized extreme value (GEV) models that include i) piecewise linear functions to model the changes in the three GEV parameters ii) adjustment coefficients for the location and scale parameters to adjust the GEV distributions of the GCM-RCM pairs with respect to the GEV distribution of the reference data. One advantage of this methodology is that it models changes in the three GEV

parameters, which makes it possible to have opposite changes between the body and the tail of the GEV distribution. For instance, at 3000 m for +2°C of global warming, we find that 100-year return levels are increasing (tail of the distribution) and that the mean annual maxima are decreasing (body of the distribution). One drawback of this methodology is that the knots of the different linear pieces ($\kappa_i$ in Eq. 1), i.e. where the slope of the piecewise linear functions changes, are fixed. The location

of these knots depends on the selected number of linear pieces (Supplement, Part F). Thus, in Figures 4-7, the amount of global warming where the slope changes (e.g. $\approx 1.6$°C, $\approx 2.1$°C, $\approx 3.2$°C) corresponds to a fixed model constraint. Another drawback of our methodology is that, some Q-Q plots display a weak fit to the tail of the distribution, which likely results from the typical high uncertainty in large quantiles (Supplement, Part B). However, a large majority Q-Q plots show a good fit, and our results are consistent when we apply statistical methodology to snowfall accumulated over 3 days and snowfall

accumulated over 5 days.

## 5.3    Results and implications

In the French Alps, previous studies on snow extremes focused on spatial patterns (Gaume et al., 2013) and past changes in spatial dependence (Nicolet et al., 2016, 2018) and marginal distribution (Le Roux et al., 2021). This study expands this knowledge to future changes in the marginal distribution of extreme snowfall.

Figure 4 and Figure 5 show that there is a consistent link between elevations and the average changes in intense snowfall, i.e. both in heavy (mean annual maxima) and extreme (100-year return levels) daily snowfall. The three types of evolution (increase, increase followed by a decrease, decrease) correspond to different outcomes of the trade-off between the projected increase in temperatures and its effect on the precipitation phase (rain vs. snow) and the projected increase in extreme winter precipitation in the French Alps (Supplement, Part D). The first type of evolution, i.e. the projected increase in intense snowfall,

probably results from the projected increase in intense winter precipitation (Supplement, Part D) and/or from the projected increase in the occurrence of optimal temperatures for extreme snowfall, i.e. temperatures located around the freezing point (O'Gorman, 2014). For elevations around 3000 m, this increase in intense snowfall is followed at +3°C by a decrease (second type of evolution), while lower elevations are directly projected to decrease (third type of evolution). For these two latter types of evolution, the decrease is likely caused by a decline in the probability of experiencing temperatures where intense

snowfall can be triggered. For elevations where a slight increase is followed by a decrease (grey curves in Fig. 4), we note that average relative changes are almost steady as a function of the global warming level between +1.5°C and +3°C. Moreover, projected changes in heavy and extreme snowfall are not substantial: average changes at +4°C of global warming range between $-15\%$ ($-11\ \text{kg m}^{-2}$) and $+8\%$ ($+13\ \text{kg m}^{-2}$) for the 100-year return levels, and between $-26\%$ ($-7\ \text{kg m}^{-2}$) and $+3\%$ ($+3\ \text{kg m}^{-2}$) for the mean annual maxima. For snowfall accumulated over 3 and 5 days, we find that trajectories of average

relatives changes remain largely similar, even if accumulation can sometimes intensify decreasing trends and temper increasing trends (Supplement, Part E).

These findings agree with IPCC (2021), which both states that "heavy snowfall events globally are not expected to decrease significantly with warming as they occur close to the water freezing point" or that "there is medium confidence that extreme snowfall events associated with winter extratropical cyclones will change little in regions where snowfall will be supported in

the future". By contrast, it must be reminded that with very high confidence, a decrease is projected for total snowfall at lower elevations for all greenhouse gas emission scenarios (IPCC, 2019). Furthermore, in general, for trends in extreme snowfall, it is difficult to understand which factor contributes the most between the thermodynamic effect and the dynamic effect that is influenced by climate variability (Faranda, 2020; Willibald et al., 2020). In the literature, trends in extreme snowfall (high return periods) are rarely quantified. Our article provides such trends as a function of global warming and elevation, and therefore supply reference values that could be assessed/compared in future studies.

More specifically, our results are comparable with the existing literature on projected extreme snowfall around the French Alps (Tab. 2). For instance, to compare with López-Moreno et al. (2011), we compute the average 25-year return level for the two time periods used in their study: 1960-1990 and 2070-2100. Specifically, for each year of the time period 1960-1990, we compute the 25-year return level using the relationship between the year and the anomaly of global mean surface temperature averaged on the six GCMs (Fig. 1). Following this method, we find that on average the 25-year return level of snowfall is projected to decrease by -15% between 1960-1990 and 2070-2100 at an elevation of 1500 m. For the same periods and elevation, López-Moreno et al. (2011) projected a decrease of -16% in the Pyrenees for the scenario SRES A2. In Table 2), we also observe that our results differ from Frei et al. (2018) but are roughly within the same range. This difference is expected as Frei et al. (2018) provide results for the whole European Alps and for elevation bands spanning 250 m elevation, while our study focuses on the French Alps for specific elevations.

| Source | Indicator | Reference period | Future period | Location | Trend |
|---|---|---|---|---|---|
| López-Moreno et al. (2011) | Average 25-year return level | 1960-1990 | 2070-2100 | Pyrenees, 1500 m | -16% |
| **Our results** | Average 25-year return level | 1960-1990 | 2070-2100 | French Alps, 1500 m | -15% |
| Frei et al. (2018) | Average Sep-May maximum | 1981-2010 | 2070-2099 | European Alps, 2250 m-2500 m | ≈ -6% |
| | | | | European Alps, 2000 m-2250 m | ≈ -9% |
| | | | | European Alps, 750 m-1000 m | ≈ -37% |
| **Our results** | Average annual maximum | 1981-2010 | 2070-2099 | French Alps, 2400 m | -17% |
| | | | | French Alps, 2100 m | -21% |
| | | | | French Alps, 900 m | -29% |

**Table 2.** Comparison of our results with the literature. In several rows of the Table (3, 4, 5), we specify that the result is approximated because the trend was read from the Figure 8 of Frei et al. (2018). Similarly to our study, Frei et al. (2018) rely on the EURO-CORDEX ensemble and scenario RCP8.5, while reported results for López-Moreno et al. (2011) rely on a single RCM driven by the scenario SRES A2.

Figure 7 illustrates that the elevation threshold, i.e. the elevation above which extreme daily snowfall is projected to increase on average with respect to +1°C of global warming, is projected to increase between +1.5°C and +4°C of global warming: from 3000 m to 3350 m for mean annual maxima, and from 2600 m to 3000 m for 100-year return levels. Thus, despite the fact that projected changes are not substantial, it needs to be verified that the design of critical infrastructures is still adequate above 2600 m of elevation, i.e. where we expect an increase in 100-year return levels on average at +1.5°C of global warming. At high elevations, changes in avalanche hazard might also entail issues in terms of road safety for high Alpine passes.

# 6 Conclusions and outlooks

This study assesses projected changes in heavy (mean annual maxima) and extreme (100-year return level) snowfall in the French Alps under the scenario RCP8.5. These changes are estimated as a function of elevation (every 300 m, from 900 m to 3600 m) and of global warming, which makes our analysis more universal than as function of time. A recent methodology (Le Roux et al., 2022) leads to new results that complement the sparse state of the art regarding the evolution of extreme snowfall events with warming in mountainous terrain. To sum up, on average, mean annual maxima is projected to decrease below 3000 m and increase above 3600 m, while 100-year return level of snowfall is projected to decrease below 2400 m and increase above 3300 m. At elevations in between, an increase is projected until +3°C of global warming, and then a decrease.

Many potential extensions of this work could be considered. First, our methodology could be applied to other regions to help understand snowfall-related hazards and anticipate how these hazards will evolve in different mountain environments. Then, our methodology could enable to analyze more in depth the seasonal trends of snowfall-related hazards. Otherwise, our study could be upgraded with modeling tools that account more explicitly for physical processes involved in the elevation-dependency of extreme snowfall trends. Finally, our study highlights the existence of a peak snowfall regime at high elevations, i.e. a global warming level (or a time) at which extreme snowfall (for a given return period) will be the largest, which was suggested theoretically by O'Gorman (2014). An extension of this work could focus even more on the characterization (timing, magnitude) of this peak snowfall regime, as it could be a key metric to anticipate future changes in snowfall-related hazards such as avalanches which also have contrasted patterns of change with elevation (Ballesteros-Cánovas et al., 2018; Giacona et al., 2021).

*Author contributions.* ELR, GE and NE designed the research. ELR performed the analysis and drafted the first version of the manuscript. RS provided the last version of the climate projection data. All authors discussed the results and edited the manuscript.

*Competing interests.* The authors declare that they have no conflict of interest.

*Code availability.* The code is publicly available at the following link: https://github.com/erwanlrx/pynonstationarygev.

*Data availability.* Information to download the data can be found in the README file of the code repository. Otherwise, the full S2M reanalysis on which this study grounds is freely available on AERIS (Vernay et al., 2019). For each GCM, the global mean surface temperature can be computed from https://climexp.knmi.nl/CMIP5/Tglobal/. For the observations, the global mean surface temperature from HadCRUT5 can be downloaded from the following webpage https://crudata.uea.ac.uk/cru/data/temperature/HadCRUT5.0Analysis_gl.txt.

*Acknowledgements.* ELR holds a PhD grant from INRAE. We are grateful to Ben Youngman for his "evgam" R package. IGE and CNRM are members of Labex OSUG. The authors also thank the editor and referees, who provided constructive and useful comments.

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
