# Peer review of "Projection of snowfall extremes in the French Alps as a function of elevation and global warming level"

_EGUsphere, 2023_

## Author Comment (AC1)

We thank the reviewer #1 for his/her very thorough comments on our manuscript. Please find below a detailed feedback to individual comments and questions. All reported typos will be accounted for in the revised manuscript.

**Major comments:**

I would appreciate a more in-depth discussion of your results. I recommend discussing your values with the results found by other authors using different snow scenarios, projections, and indicators. You should also discuss the seasonality of snowfall, the physical reasons for the changes you detected, the implications of your results, and why it is important to analyze extreme snowfall. A thorough review of the results found by other authors in the Alps, Pyrenees, Apennines, and nearby mountain ranges could complete the study and make it more relevant.

In the discussion of the results we already mention some "physical reasons" for the three types of changes we highlight, and the implications in terms of adaptation of critical infrastructures. This mandatory adaptation to snowfall hazard (as mentioned in the beginning of the introduction) explains why studying changes in extreme snowfall is important. Changes in seasonality in annual maximum snowfall are not included in our study as we believe that it is a study on its own and the paper is already long, and will be increased by the analysis of maximum 3-day snowfall. The revised version of the paper will mention that this should be done in further studies. In the discussion of the results, we will add a comparison of our values (extracted for specific time windows) with two other articles in the region: in the Alps (Frei et al. 2018) and in the Pyrenees (Lopez-Moreno et al. 2011).

In the Introduction we will add that "Many recent studies based on climate projections illustrate this phenomenon with maps of extreme snowfall trends (Lader et al. 2020, Kawase et al. 2021, Chen et al. 2020, Quante et al. 2021). For instance, Lader et al. 2020 found that maximum 2-day snowfall projections either show no trend or decreasing trend for RCP8.5. On the other hand, increasing trends in annual maximum and monthly maximum daily snowfall have been found over several regions in Japan for RCP8.5 (see Fig. S3 of Kawase et al. 2021). In the Northern Hemisphere, Chen et al. 2020 finds significant increasing trends in high-latitude regions, and Quante et al. 2021 show an intensification of extreme snowfall across large areas." However, we believe that a full comprehensive review of (extreme and non extreme) past and future snowpack trends is out of scope: such thorough review could likely result in an article on its own.

**Comments:**

**Abstract:**

L8: Please specify the duration (months) of the season you analyzed.

We will reorganize the sentences in the abstract to mention that we study annual maxima in the hydrological sense (from August 1st to July 31st).

L8 – L9: I suggest briefly showing the projected changes in temperature and precipitation under RCP8.5.

We believe that this information should not be mentioned in the abstract because it is not a specific result of the article. However, global temperature changes under RCP8.5 scenario are provided in Figure 1 c), and global precipitation trends with global warming are mentioned in the introduction. In addition, trends in extreme precipitation in the French Alps under RCP8.5 are provided in Appendix, which supports the discussion regarding the explanation of the changes we highlight in extreme snowfall.

L10: Please specify what massifs are (i.e., mountain zones with similar meteorological conditions) when you mention "available massifs."

In the revised manuscript, we will remove "available" from this sentence to keep the abstract simple. The fact that the average is not always computed on all massifs is detailed in our methodology: "Then, average relative change is defined as the relative change averaged over all available massifs of the French Alps. Massifs are considered as not available when the considered elevation is above the top elevation of the massif, or when the frequency of years without snowfall is above a threshold because it might break extreme value theory assumptions."

L12: Please briefly show the spatial differences found (if any) in your study.

Spatial differences are mentioned in the Result section (L126-L131). For the 100-year return levels, we did not find any striking spatial pattern. For the mean annual maxima, we often observe a typical North-South pattern but not always. Therefore, to not sound overconfident and since this contribution is not decisive, we prefer to not mention this result in the abstract, which focuses on the main robust result.

L14: Please avoid mentioning absolute values here.

In the revised manuscript, we will remove absolute values from the abstract.

L15: Please include the statement "Finally, for each global warming level..." between L5 and L8, where you briefly explain the methods.

We will add this statement in the revised manuscript.

L16: "The elevation threshold is projected to rise between +1.5∘C and +4ºC: from 3000 m to 3350 m" to address the question. There are no changes if warming is <= 1.5ºC ? and for > 4ºC?, please rephrase.

In the revised manuscript, we will reformulate this sentence and the previous sentence as "Finally, for each global warming level between +1.5ºC and +4ºC, we compute the elevation threshold that separates contrasted trends, i.e. where the average relative change equals zero. This elevation threshold is projected to rise from 3000 m at +1.5ºC to 3350 m at +4ºC for mean annual maxima, and from 2600 m to 3000 m for 100-year return levels. These results have implications for the management of risks related to extreme snowfall."

**Introduction**

L35: I recommend describing the main conclusions of these works.

These works on projected extreme snowfall provide results over gridded areas which are either too local/regional or too broad/global. Therefore, we decided initially not to describe them in detail since they do not provide relevant information for our study. To follow the reviewer recommendation, in the revised manuscript we will add that:

"Many recent studies based on climate projections illustrate this phenomenon with maps of extreme snowfall trends (Lader et al. 2020, Kawase et al. 2021, Chen et al. 2020, Quante et al. 2021). For instance, Lader et al. 2020 found that maximum 2-day snowfall projections either show no trend or decreasing trend for RCP8.5. On the other hand, increasing trends in annual maximum and monthly maximum daily snowfall have been found over several regions in Japan for RCP8.5 (see Fig. S3 of Kawase et al. 2021). In the Northern Hemisphere, Chen et al. 2020 finds significant increasing trends in high-latitude regions, and Quante et al. 2021 show an intensification of extreme snowfall across large areas."

L49: A better justification is needed to highlight the innovative aspects of this study, distinguishing it from previous works, since a similar degree approach methodology has been previously applied (such as Verfaille et al. (2017) in The Cryosphere).

Verfaillie et al. (2018, The Cryosphere) did not study extreme snowfall projections. In the revised manuscript, to emphasize the innovative aspects of this study, we will add that "The innovative aspect of our study is twofold: i) the novelty of the statistical methodology for extreme snowfall based on non-stationary GEV distribution and GMST as a covariate ii) trends for projected extreme snowfall in the French Alps."

Considering that the introduction, Figure 2, and Table 1 are similar to LeRoux et al. (2021), I suggest providing a more comprehensive review of past and future snow trends in mid-latitude mountain areas, including complementing extreme snowfall trends with other snowpack projections. The questions that this study aims to answer should be introduced in the last sentences of the introduction.

Table 1 is similar to Le Roux et al. (2021) in its structure, but differs in its contents: Le Roux et al. (2021) focused on past extreme snowfall trends, while this article focuses on future extreme snowfall trends. As mentioned in a previous comment, in the Introduction we will add that "Many recent studies based on climate projections illustrate this phenomenon with maps of extreme snowfall trends (Lader et al. 2020, Kawase et al. 2021, Chen et al. 2020, Quante et al. 2021). For instance, Lader et al. 2020 found that maximum 2-day snowfall projections either show no trend or decreasing trend for RCP8.5. On the other hand, increasing trends in annual maximum and monthly maximum daily snowfall have been found over several regions in Japan for RCP8.5 (see Fig. S3 of Kawase et al. 2021). In the Northern Hemisphere, Chen et al. 2020 finds significant increasing trends in high-latitude regions, and Quante et al. 2021 show an intensification of extreme snowfall across large areas." However, we believe that a full comprehensive review of (extreme and non extreme) past and future snowpack trends is out of scope: such thorough review could likely result in an article on its own.

We believe that the question this study answers is clear based on our title and the last paragraph of our introduction "Our study assesses projected changes in heavy (mean annual maxima) and extreme (100-year return level) snowfall in the French Alps under a high emission scenario (RCP8.5).". Thus, we believe it might be too redundant to add a sentence such as "What are the projected trends of snowfall extremes in the French Alps ?". However, as mentioned in a previous comment, to better emphasize the novelty of our work, in the revised manuscript we will add at the end of the introduction that "The innovative aspect of our study is twofold: i) the novelty of the statistical methodology for extreme snowfall based on non-stationary GEV distribution and GMST as a covariate ii) trends for projected extreme snowfall in the French Alps."

**Data and methods**

L57. A detailed explanation is needed to clarify why the S2M works by 300 m and massifs.

The SAFRAN reanalysis, used for the SAFRAN - SURFEX/ISBA-Crocus - MEPRA, has been developed since its inception in the early 1990s on the basis of a compromise between the spatial density of in-situ observations in mountain areas, the spatial variability of meteorological conditions in mountain areas, both horizontally and vertically (as a function of elevation). The choice of 300m-spaced elevation bands, over massifs considered to be homogeneous for a given elevation, stems from these initial investigations, to which further work did not show that it would be adequate to change the vertical resolution of the SAFRAN reanalysis. For more information, see Durand et al., (1993) and Durand et al. (2009a).

The ADAMONT methods should be described more comprehensively, including the weather types included and the rationale for selecting quantile-mapping.

In the Data section, we will add the following explanation for the ADAMONT method:

"Annual maxima of daily snowfall are computed in three steps. First, daily precipitation from the EURO-CORDEX project (Jacob et al, 2014) are downscaled and corrected at the massif-level and every 300 m of elevation using an advanced quantile-mapping method (so called ADAMONT, see Verfaillie et al. 2017). This ADAMONT method relies on the S2M reanalysis as reference and is applied separately on four seasons and four weather regimes. Then, daily precipitation is disaggregated at the hourly time step using analogues and partitioned between rain and snow with the threshold 1°C and an additional quantile-mapping correction is applied (see Verfaillie et al., 2017, for details). Finally, daily snowfall are computed by aggregating hourly snowfall and the annual maxima of the daily snowfall values are computed (annual values taken from August to July, so as to cover the hydrological year)"

The quantile-mapping was selected because it is a standard and robust debiasing method which is commonly used (Maraun et al 2017).

**Results**

The article's content needs further editing in several places. For instance, the description of the data and methods should avoid using the term "observations" since the data analyzed are not observed data (L103). Sections 4.1 and 4.2 should describe the maximum and minimum changes for both indicators, as well as any seasonal and spatial differences, and provide an explanation for why the study focuses on 100-year return levels (L124).

In the revised manuscript, we will replace the term "past observations" with "reference data" for clarity. We will also add a description of maximum and minimum changes for both indicators. Changes in seasonality in annual maximum snowfall are not included in our study as we believe that it is a study on its own and the paper is already long, and will be increased by the analysis of maximum 3-day snowfall. The revised version of the paper will mention that this should be done in further studies. Spatial différences/patterns are already analyzed in Sections 4.1. The study focuses on 100-year return levels because this return period is often considered in snow-related hazard study. We will add a sentence in the introduction to explain this choice.

L140: The article could be improved by showing the monthly and spatial differences in the results, including whether the increases at high elevations are expected for winter and whether these trends are consistent across seasons. Finally, there are several lines where you mentioned specific massifs (i.e., Mercantour massif). For a non-local reader, it can be challenging to follow.

Changes in seasonality in annual maximum snowfall are not included in our study as we believe that it is a study on its own and the paper is already long, and will be increased by the analysis of maximum 3-day snowfall. The revised version of the paper will mention that this should be done in further studies. However, as shown in the Appendix, we provide an analysis of winter precipitation annual maxima. Yes we will replace "Mercantour massif" by " the most south eastern massif (Mercantour massif)" and "Vanoise massif" by "a large north eastern massif (Vanoise massif)" to make the text more accessible to a non-local reader.

In the methodology section, you described the S2M process as occurring in 300-meter increments. However, at a certain point, you changed to four single elevations (as shown in Figures 2 and 3). Could you please provide an explanation for this change and ensure consistency in the methodology?. A statement in the methodology section or results would be appreciated.

Yes the S2M data is available every 300 m from elevation 900 m to 3600 m. Thus, in total we have ten elevations, but only show four single elevations (the two extremal elevations 900 m, 3600 m and two intermediate elevations 1800 m et 2700 m) due to display constraints. We will add a sentence at the beginning of the concerned subsection to clarify that in this subsection we only display a subset of elevations.

Figure 3, it would be beneficial to include the values inside the massifs.

In the revised version of the manuscript, we will add values for Figure 2 and Figure 3.

**Discussion: Data and methods, section 5.1 and 5.2.**

It is recommended that you acknowledge the limitations of RCMs, quantile mapping, and the lack of observations in mountainous regions. It would be essential to better discuss how you overcame these limitations, as well as any differences between previous extreme values metrics. I reccomend to provide more information on the ADAMONT method and the spatial differences, such as the number of weather types categorized (and if the bias correction was the same all massifs and months). Additionally, you used quantile mapping for statistical adjustment and an analogous approach for sub-daily disaggregation. It is suggested that you at least mention the limitations and uncertainties of a simple quantile mapping approach. Please see a review by Maraun et al. (2017) in Nature Climate Change.

In the revised manuscript we will add:

- For the lack of observations in mountainous regions that "Above 2700 m, a few stations record surface temperatures and daily precipitation amounts. Snowfall amounts at high elevations are thus more uncertain, and an underestimation of precipitation above 3000 m can be suspected (Vionnet et al. 2019). Annual maxima of daily snowfall are likely affected by these uncertainties and by the high spatial and temporal variability of annual maxima from daily values. Despite these known limitations, this reanalysis has been repeatedly shown to provide valuable insights regarding past snow climate conditions and their links to snow avalanche activity in the French Alps (Durand et al., 2009; Castebrunet et al., 2012; Schläppy et al., 2016).".

- That quantile mapping has known limitations: they can sometimes lead to an implausible climate and they cannot overcome climate model errors (Maraun et al. 2017). However, in ADAMONT quantile mapping is applied for quantiles below 0.995: otherwise "a constant adjustment based on the value of this last quantile is applied in order to allow for new extremes"

- For RCMs, the main limitation is that they do not explicitly model convection and thus they fail to simulate convection based precipitation. This limitation might particularly have an impact on precipitation in the Southernmost massifs of the French Alps where convective processes play a major role. However, convective phenomena occur mostly in summertime, and most extreme precipitation events occur in wintertime throughout the mountain regions.

Since one of the principal conclusions of this study is the projected increase in extreme snowfall at high elevations, the manuscript can be enhanced by including a section where the authors describe the limitations of the S2M reanalysis and how it could impact the obtained results. The authors should further explain the differences in irradiance as reported by Quéno et al. (2017), precipitation (Vernay et al., 2019) or snow (Vionnett et al., 2019). Additionally, the differences between high-resolution non-hydrostatic models used in other studies, such as Musselman et al. (2017) in Nature, could be discussed.

We mention in the discussion that "strong uncertainties remain at high elevations" for the S2M reanalysis, its most important limitation being probably the lack of in-situ observations assimilated at high elevations. Issues with SAFRAN incoming radiation and with snow cover simulations are not relevant to our study on extreme precipitation, which exploits only the extreme precipitation fields from the SAFRAN reanalysis. Limitations of the S2M reanalysis are mentioned in the manuscript, and we would like to insist here on the fact that our study does not only provide results based on climate projections adjusted using S2M, but primarily introduce and illustrate a novel methodological framework, which can be applied to upcoming datasets, which will hopefully suffer from less limitations.

**Discussion: Results, section 5.3**

It is recommended to expand the discussion and compare the results with previous studies. There are relevant differences between snow indicator (i., snowfall and snow melt rate) see Musselman et al. (2017) in Nature that will be worth to include.

As mentioned in the major comments, we will add a comparison of our values (extracted for specific time windows) with two other articles in the region: in the Alps (Frei et al. 2018) and in the Pyrenees (Lopez-Moreno et al. 2011). We will not mention snow melt rate because snowpack variables are out of the scope of the current study.

In addition, the authors should briefly describe previous findings in the Alps, such as those reported by Marty et al. (2017) in The Cryosphere, Piazza et al. (2014) in Climatic Change, Terzago et al. (2017) in The Cryosphere, and Steger et al. (2014) in Climate Dynamics, among others. The authors should also address precipitation seasonality, as presented by Kotlarski et al. (2022) in Climate Dynamics, and explain how snow changes are exceptional within a long-term perspective, as shown by Carrer et al. (2023) in Nature Climate Change. Differences between CMIP 5 and 6 and by scenario should be noted. The authors should mention how recent increases in winter extreme snowfall events counterbalanced summer glacier ablation in Italian alpine regions, as reported by Colucci et al. (2021) in Water.

As explained in a previous commentary, we believe that our article provides a broad review of the results found in the literature on future extreme snowfall trends. However, most proposed articles do not seem to focus on snowfall but on snowpack variables

- Marty et al 2017 focuses on snow depth.
- Terzago et al 2017 focuses on snow depth.
- Steger et al 2014 focuses on mean winter snow water equivalent.
- Kotlarski et al 2022 focuses on snow cover (snow cover duration, snow depth).
- Carrer et al 2023 focuses on snow cover duration
- Colucci et al 2021 focuses on snow dover (snow depth, snow total accumulation, snow thickness

with the exception of:

- Piazza et al 2014 that focuses on total snowfall. However, in this article, we did not find any results/analysis that was relevant for our study.

L215: You can include a review by Faranda et al. (2020) in Weather and Climate Dynamics, and references therein, who discuss recent snowfall changes due to thermodynamics and changes in the atmospheric circulation..

L225: It is therefore essential to acknowledge previous research on natural climate variability at high elevations in the Swiss Alps, such as the work of Willibald et al. (2020) in The Cryosphere and the references therein.

Faranda 2020 is not a review, but an analysis of recent trends in annual maxima of snowfall for Europe at a large scale (NUTS-0 and NUTS-2 levels). We cited this article in Le Roux et al 2021, which focuses on past extreme snowfall. On the other hand, Willibald focuses on the impact of climate variability on mean winter snow depth. To acknowledge these previous works, in the revised version of our manuscript, we will cite both articles in the discussion of the result to note that "For trends in extreme snowfall, it is difficult to understand which factor contributes the most between the thermodynamic effect and the dynamic effect that is influenced by climate variability (Faranda 2020, Willibald et al. 2020)".

L232: Additionally, it is highly recommended that the potential environmental and social impacts of such projections be discussed.

At the end of the discussion, we mention that "it needs to be verified that the design of critical infrastructures are still adequate above 2600 m of elevation, in particular with respect to avalanche hazard and mountain buildings (e.g., due to the accumulation of snow on roofs)". At high elevation, changes in avalanche hazard might also entail issues in terms of road safety for high Alpine pass.

L237: It is advised to remove quotes from the conclusion section. Finally, to promote transparency and reproducibility, it is recommended to include a statement on the availability of data and code.

Following the advice, we will remove the quotes around "peak snowfall regime". We will also add "code availability" and "data availability" sections after the appendix section. Our code is available on Github. Instructions to download the data and run the code are available in the README file of the Github project.

---

## Author Comment (AC2)

We thank the reviewer #1 for his/her thorough comments on our manuscript. Please find below a detailed feedback to individual comments and questions. All reported typos will be accounted for in the revised manuscript.

**Major points**

Methodology in general is lacking a lot of key information:

- Did you take the snowfall variable directly from the RCMs or how?
- What exactly was ADAMONT used for and how?
- What did you use S2M for: as past reference, or only in the bias adjustment? It seems like you concatenated S2M with a GCM-RCM, but please correct me if I'm wrong. And if yes, why did you not use the historical data of the GCM-RCM instead? I would expect some kind of break when merging a reanalysis and a climate model.

These three questions were answered in the second paragraph of our Data section. In the revised manuscript, we will add the following explanation:

"Annual maxima of daily snowfall are computed in three steps. First, daily precipitation from the EURO-CORDEX project (Jacob et al, 2014) are downscaled and corrected at the massif-level and every 300 m of elevation using an advanced quantile-mapping method (so called ADAMONT, see Verfaillie et al. 2017). This ADAMONT method relies on the S2M reanalysis as reference and is applied separately on four seasons and four weather regimes. Then, daily precipitation is disaggregated at the hourly time step using analogues and partitioned between rain and snow with the threshold 1°C and an additional quantile-mapping correction is applied (see Verfaillie et al., 2017, for details). Finally, daily snowfall are computed by aggregating hourly snowfall and the annual maxima of the daily snowfall values are computed (annual values taken from August to July, so as to cover the hydrological year)"

Therefore to answer the initial questions of the referee: We did not concatenate S2M with GCM-RCM. For each GCM-RCM we consider both historical data and the data for the scenario RCP8.5. We do not consider the snowfall variable from the RCMs, but the precipitation from the RCM. First, this precipitation is downscaled at the massif-level and every 300 m with the ADAMONT method with the precipitation from S2M as reference. Then, hourly precipitation data from ADAMONT is partitioned into rainfall and snowfall using a threshold at 1 degree celsius for the transition from rain to snow. Finally, annual maxima of daily snowfall are computed. But in the historical period these annual maxima differed with annual maxima of the S2M reanalysis, because the chronology of meteorological events differs between climate model simulations (RCM) and the S2M reanalysis. This is the reason why in our non-stationary GEV model we rely on annual maxima of snowfall from S2M as reference.

- I would have expected a few more references on GEV in your description. For readers it would be useful to have an overview paper for general non-stationary GEV, which is what most people use. Then an adaptation to GEV with covariates (what you call non-stationary, but in principle this works with any covariate) and has been developed years ago.

Yes we agree, and do not claim to have developed the adaptation of GEV to covariates. For clarity, and for readers of the Cryosphere, our objective was to present our non-stationary GEV model as concisely as possible, and point interested readers to our previous article for a more thorough understanding of the non-stationary GEV model. In the methodology, we will add a citation to Coles 2001 (and reference therein) which is a reference book for non-stationary GEV. In climatology non-stationary GEV have been used extensively, e.g. (Maraun et al. 2009). The originality of our methodology is to consider a non-stationary GEV model fitted both on GCM-RCM pairs and reference data.

- Your application to GCM-RCM pairs seems like a prime example of using a random effects specification to reduce the number of parameters while accounting for model variability and interdependence. I'm not up-to-date on recent GEV literature and developments, but I guess there should be some random effects GEV models (not necessarily in your specific field). Would be nice to see this as a discussion point at least.

Yes, that's exactly the point of our GEV model. In the introduction and the discussion of Le Roux et al. 2022, we reported articles in climatology that are using similar techniques (additional and/or multiplicative random effects) for GCM-RCM ensemble. For instance, Brown et al 2014 estimated a non-stationary GEV distribution with observations and a single GCM-RCM and introduced additive coefficients for each GEV parameter. Other authors considered multiplicative coefficients (Hosseinzadehtalaei et al., 2021). To the best of our knowledge, we do not know any comparable GEV models outside of climatology. The idea to rely on random effects stems from the ANOVA framework, which have been applied to partition the uncertainty of GCM-RCM projections (Hawkins and Sutton, 2009; Evin et al., 2019).

- Since almost no natural physical process follows a piecewise linear function with abrupt breakpoints, I wonder why you chose this method? What is the benefit to more flexible approaches like a spline basis, which has a similar number of parameters but does not depend on choosing breakpoints and number of breakpoints (which you note as limitation in the discussion)? Then, even if you did some validation in your snow load paper, I would expect a goodness-of-fit test also here, since snowfall might behave differently than snow load. So as to justify why a piecewise linear function is fitting better than a simple linear model (or one without covariates, by-the-way).

In the literature, most non-stationary GEV models rely on linear functions of the covariate. Thus, such models cannot model a more complex trajectory, e.g. increase of mean maximum followed by a decrease in mean maximum. In order to allow a modeling of more complex behavior, while having as few parameters as possible, we decided to rely on piecewise linear functions, which is the same as degree 1 splines. In other words, we choose to rely on the most simple function that is capable of approximating more complex behavior. We will add goodness-of-fit tests with the selected models, simple linear models and constant models.

- Have you assessed the shape parameter of your GEV functions? The shape parameter decides on Gumbel, Weibull, or Frechet type, and each of these have a very different behaviour on tails and expectance of extremes, and your use of adjustment parameters might change the type of GEV in the future – so I was wondering whether you have analyzed this also?

No, we did not analyze this parameter. We did not display this parameter because our objective was to provide an article accessible to readers that do not necessarily have a strong background in statistics, and because this parameter can be difficult to interpret. In the revised manuscript, we will add in the Result section sentences on the shape parameters based on Figures (maps and/or trajectory of the shape parameter with GMST) that will be located in the Supplementary Materials.

Annual maxima from daily values (? I guess, this is not specified in the manuscript) are notoriously unstable. This deserves some discussion on the challenges in using annual maxima of snowfall, which already suffers from high spatial and temporal variability.

In the Discussion section on the data, we will add the following sentence "Snowfall amounts at high elevations are thus more uncertain, and an underestimation of precipitation above 3000 m can be suspected (Vionnet et al, 2019). Annual maxima of daily snowfall are likely affected by these uncertainties and by the high spatial and temporal variability of annual maxima from daily values."

Finally, from a user and climatological point of view, I would say that besides maximum 1-day snowfall, 3-day and 5-day snowfall are of particular interest, even more than the 1-day one. In fact, many assessments of future climate change focus not only on 1day annual maxima but also on 3, and 5 day precipitation extremes (cf. Rx*day indices from ETCCDI). This is beneficial in multiple ways: First it does not introduce an arbitrary break at time=0 for the accumulation of daily values. Second, it also considers changes in circulation regimes, such as increased persistence. Third, problematic recent snowfall extremes in the Alps were often multi-day events. This study could benefit strongly from also including 3 and 5 day extremes in the analysis, making the results both scientifically more robust and with more societal relevance.

We agree with this remark. We will include in the revised manuscript a brief analysis of maximum 3-day snowfall which is, e.g. the most relevant variable for avalanche risk assessment.

**Minor points:**

- L58: I suggest not calling a reanalysis an "observation", not even for simplicity. This can be very misleading for a reader who does not carefully read every detail.

In the revised manuscript, we will remove this sentence. In the rest of this article, we will replace the term "past observations" with "reference data".

- Fig1c: Smoothed, how exactly? Please indicate.

In the revised manuscript, we will add in the Data Section of the revised manuscript that smoothing was applied using cubic splines.

- L127: missing sentence?

In the revised manuscript, we will reformulate this sentence as "For the mean annual maxima, relative changes are often larger (increasing trends are greater, decreasing trends are less marked) in the Northern French Alps than in the Southern French Alps."

- L129: -40% is not a substantial change?

We will remove this sentence in the revised manuscript. This sentence was only meant to explain that changes were not as high as we expected.

- L205: "huge step forward" is not a justified conclusion. Just using a more complicated model does not automatically imply better results. To justify this sentence, I'd expect some tests of goodness-of-fit (see also related comment above).

In the revised manuscript, we will remove the sentence mentioning a "huge step forward" from the discussion. Yes, as explained above, we will add some goodness-of-fit.

References:

Maraun et al. 2009, The annual cycle of heavy precipitation across the United Kingdom: a model based on extreme value statistics, https://doi.org/10.1002/joc.1811

Hawkins, E. and Sutton, R. 2009, The potential to narrow uncertainty in regional climate predictions, https://doi.org/10.1175/2009BAMS2607.1

Brown, S. et al. 2014  Climate projections of future extreme events accounting for modelling uncertainties and historical simulation biases, https://doi.org/10.1007/s00382-014-2080-1

Evin, G et al. 2019, Partitioning Uncertainty Components of an Incomplete Ensemble of Climate Projections Using Data Augmentation, https://doi.org/10.1175/JCLI-D-18-0606.1

Hosseinzadehtalaei, P et al. 2021, Climate change impact assessment on pluvial flooding using a distribution-based bias correction of regional climate model simulations, https://doi.org/10.1016/j.jhydrol.2021.126239

---

## Author Response (AR2)

**Point by point reply to the comments**

Dear Editor,

We thank you and the referees for their thorough reviews and for the suggestions.

Please find below, point by point, our answers to all the suggestions. Your suggestions are in red, suggestions from Referee #2 are in blue.

Yours sincerely,

On behalf of the co-authors

Erwan Le Roux

My concern is your conclusions and outlook section. In principle, the first paragraph summarizes the applied method, whereas the second one gives some outlook. Could you please add some information on the major results or achievements?

In the conclusion, at the end of the first paragraph, we added our main result: "To sum up, on average, mean annual maxima is projected to decrease below 3000 m and increase above 3600 m, while 100-year return level of snowfall is projected to decrease below 2400 m and increase above 3300 m. At elevations in between, an increase is projected until +3°C of global warming, and then a decrease."

L75: Since you partition snowfall with a temperature threshold, I guess you also need to downscale temperature with ADAMONT, and not only precip?

Yes, we added a sentence to clarify this point in the Data Section. "The ADAMONT method relies on the S2M reanalysis as a reference and is applied separately for the four seasons, four weather regimes, and several meteorological variables including temperature and precipitation."

L80: Please explain why you don't use directly the daily RCM data and instead you first disaggregate daily to hourly, and then calculate daily from hourly. I can imagine a few reasons (e.g., related to the daily cycle, etc.). But then disaggregating precip is not a trivial task (compared to temperature, where you go fairly well with sine-cosine functions). Have you tested uncertainty or sensitivity to this methodological step in some previous study? If not, please at least mention and/or discuss it.

Yes, as mentioned in the Data Section, "daily precipitation is disaggregated at the hourly time step using analogues and partitioned between rain and snow with the threshold 1°C and an additional quantile-mapping correction is applied". The reason for this disaggregation is that hourly-data is needed to force energy balance land surface model. The ADAMONT method has been analyzed in the original publication, see Verfaillie et al., 2017. Moreover, the ADAMONT method has already been used at the European-scale, mainly in mountainous areas, using UERRA as reference (Morin et al., 2021, Climate Services).

References:

Verfaillie et al. 2017, The method ADAMONT v1.0 for statistical adjustment of climate projections applicable to energy balance land surface models, https://doi.org/10.5194/gmd-10-4257-2017

Morin et al. 2021, Pan-European meteorological and snow indicators of climate change impact on ski tourism, https://doi.org/10.1016/j.cliser.2021.100215